# A Weighted Facial Expression Analysis for Pain Level Estimation

**DOI:** 10.3390/jimaging11050151

**Published:** 2025-05-09

**Authors:** Parkpoom Chaisiriprasert, Nattapat Patchsuwan

**Affiliations:** College of Digital Innovation Technology, Rangsit University, Pathumthani 12000, Thailand; nattapat.p67@rsu.ac.th

**Keywords:** pain level, facial expressions, weighting method, action unit

## Abstract

Accurate assessment of pain intensity is critical, particularly for patients who are unable to verbally express their discomfort. This study proposes a novel weighted analytical framework that integrates facial expression analysis through action units (AUs) with a facial feature-based weighting mechanism to enhance the estimation of pain intensity. The proposed method was evaluated on a dataset comprising 4084 facial images from 25 individuals and demonstrated an average accuracy of 92.72% using the weighted pain level estimation model, in contrast to 83.37% achieved using conventional approaches. The observed improvements are primarily attributed to the strategic utilization of AU zones and expression-based weighting, which enable more precise differentiation between pain-related and non-pain-related facial movements. These findings underscore the efficacy of the proposed model in enhancing the accuracy and reliability of automated pain detection, especially in contexts where verbal communication is impaired or absent.

## 1. Introduction

Pain is a subjective experience that cannot be directly perceived or observed by others. Consequently, healthcare professionals often encounter challenges in accurately assessing a patient’s pain severity due to its inherently personal and introspective nature. In many cases, patients may struggle to articulate their pain clearly or may be reluctant to do so, thereby complicating the processes of diagnosis and treatment planning [1]. In situations where patients are unable to communicate verbally, facial expressions often serve as a primary medium for conveying discomfort [2]. These scenarios underscore the critical importance of non-verbal pain assessment techniques in facilitating effective and timely medical intervention. This study aimed to develop and validate a novel weighted facial expression analysis model specifically designed to improve the accuracy and robustness of automated pain intensity estimation for non-verbal patients.

Contemporary research has primarily explored two approaches for automated pain assessment. The first approach involves analyzing facial video sequences to identify real-time expressions, which are then compared with annotated datasets to estimate pain severity levels [3,4,5]. This technique aligns detected facial movements with the existing labeled data to produce pain intensity evaluations. The second approach focuses on assessing facial muscle activity through the analysis of standardized action units (AUs), as defined by the facial action coding system (FACS), a widely recognized framework for decoding facial expressions [6]. A comprehensive metric frequently employed in this context is the Prkachin and Solomon pain intensity (PSPI) scale, which quantifies pain through specific AU activations [7]. Automated systems have been developed to predict PSPI scores on a frame-by-frame basis, thereby enhancing the objectivity and efficiency of pain evaluation.

Despite notable advancements, the existing methodologies continue to face limitations, particularly when interpreting subtle or ambiguous facial expressions influenced by contextual variables. To address these challenges, the present study introduced a novel weighting mechanism that integrates facial expression recognition (FER) with AU-based analysis. By incorporating expression-driven weight adjustments into AU processing, the proposed method aimed to significantly improve the precision and robustness of pain intensity estimation.

This research began with a comprehensive review of the existing literature and methodologies related to automated pain assessment, forming the conceptual foundation for the proposed approach. The method employs an innovative fusion of FER techniques [8] and AU analysis to achieve more accurate pain level estimation. Its effectiveness was empirically validated using a dataset comprising 4084 facial images from 25 individuals, demonstrating superior accuracy compared to traditional estimation methods.

The remainder of this paper is structured as follows: Section 2 details the research methodology, including dataset preparation, normalization procedures, face mesh detection using MediaPipe, and AU zone computation. Section 3 presents the experimental results along with a comparative analysis against baseline approaches. Section 4 discusses the implications of the findings, highlights the strengths and limitations of the proposed method, and outlines avenues for future research, including the incorporation of multimodal data and the expansion of dataset diversity. Finally, Section 5 concludes the paper by summarizing key findings and proposing directions for continued investigation.

## 2. Related Work

### 2.1. Facial Expression Analysis for Pain Detection

Facial expression analysis has emerged as a pivotal technique in the assessment of pain severity, with ongoing research efforts directed toward improving the accuracy and reliability of this approach. Among the early contributors to this field, Manolova and Neshov [1] employed facial landmark detection in conjunction with the supervised descent method (SDM) to automate pain level estimation. One of the key limitations identified in their work—and in the field more broadly—was the scarcity of high-quality training data. Addressing this issue, ref. [7] focused on the detection of shoulder pain through facial recordings obtained during both passive and active physical movements. Their study emphasized the critical role of robust, well-annotated datasets in enhancing model performance and generalizability.

Building upon these foundational studies, Xiaojing Xu and Virginia R. de Sa [3] introduced uncertainty estimation into video-based facial pain recognition, systematically evaluating a range of model architectures to identify those that optimized diagnostic accuracy. In a subsequent study, ref. [8] further advanced the domain by incorporating multitask learning frameworks to predict pain scores using a visual analog scale. This approach involved aggregating frame-level observations into a multidimensional scoring system, thereby improving the granularity and interpretability of automated pain assessments.

Collectively, these contributions have significantly advanced the field of facial expression-based pain detection, addressing core methodological challenges and establishing a foundation for the development of more sophisticated, data-driven diagnostic tools in clinical settings.

### 2.2. Neural Networks and Deep Learning for Pain Detection

Neural networks and deep learning models have significantly advanced the field of facial expression-based pain detection by improving the accuracy and robustness of pain intensity estimation. A notable contribution in this domain is presented in [9], where recurrent neural networks (RNNs) trained on sequences of facial expressions were employed to predict pain intensity. This study highlighted the effectiveness of temporal modeling in capturing dynamic facial cues associated with pain, underscoring the value of sequential data analysis in medical diagnostics.

Expanding upon this approach, Shier, W.A. and Wilkinson, S. [10] introduced a hybrid model that integrates RNNs with hidden conditional random fields (HCRFs) to further enhance pain intensity estimation. This integration allowed for the exploitation of both temporal dependencies and probabilistic modeling, enabling more precise predictions of pain levels based on the visual analog scale (VAS). This method marked a significant improvement over traditional techniques by addressing the challenges posed by variations in facial expressions and context-specific cues.

In parallel, Shier and Yanushkevich [5] pursued an alternative methodological direction by examining the efficacy of handcrafted feature extraction in combination with classical machine learning algorithms. Utilizing Gabor energy filters and support vector machines (SVMs), their system was able to classify pain severity into distinct categories: mild, moderate, and severe. This approach demonstrated that, despite the rise of deep learning, traditional techniques remain valuable when paired with effective feature representation strategies.

Together, these studies illustrate the evolving landscape of computational pain assessment, highlighting the complementary strengths of deep learning architectures, probabilistic models, and conventional machine learning techniques. The continued refinement of these methodologies holds substantial promise for improving the objectivity, interpretability, and clinical applicability of automated pain detection systems.

### 2.3. Facial Action Coding System (FACS) and Emotion Recognition in Pain

Facial action unit analysis has emerged as a fundamental approach for assessing pain, particularly in patients who are unable to verbally communicate their discomfort. The foundational work [11] led by Zhanli Chen and Guglielmo Menchetti explored the identification of specific facial muscle movements associated with pain using the facial action coding system (FACS). Their study emphasized the importance of recognizing subtle facial expressions, which can serve as critical indicators of pain in non-communicative individuals. By accurately detecting these nuanced muscular changes, their research contributed to the refinement of automated pain recognition systems in clinical settings.

Building on this foundation, ref. [12] proposed a more advanced two-step framework aimed at enhancing the accuracy of pain detection. This method begins by identifying AUs at the individual frame level and subsequently applies multiple instance learning (MIL) techniques to analyze sequences of frames across a video. This temporal integration enables a more holistic and robust assessment of pain, accounting for fluctuations in expressions that may occur over time.

Complementing AU-based analysis, ref. [4] introduced a geometric computation-based technique that leverages facial landmarks to classify pain-related expressions. Utilizing the k-nearest neighbors (k-NN) algorithm, this approach effectively distinguished between pain and neutral expressions, demonstrating the potential of classical machine learning techniques in this context. By quantifying geometric relationships between facial points, the model offered an interpretable and computationally efficient solution for real-time pain assessment.

Collectively, these studies underscore the increasing importance of AU-focused methodologies and geometric modeling in the development of automated pain detection systems. They highlight the value of combining temporal dynamics, machine learning, and detailed facial analysis to support more accurate, accessible, and non-invasive pain assessment tools, particularly for patients with limited communicative abilities.

### 2.4. Emotional Expression and Pain Detection in Special Populations

Facial expression-based pain detection plays an increasingly critical role in assessing pain among special populations, particularly individuals with communicative limitations due to neurological disorders or developmental stages. Patients with neurodegenerative conditions, such as Parkinson’s disease, often exhibit reduced facial expressivity, complicating conventional pain assessment. Addressing this issue, ref. [13] investigated the emotional facial expressions of Parkinson’s patients compared to healthy individuals, revealing notable differences in expression intensity. These findings emphasize the need for specialized assessment strategies that account for altered facial dynamics in such populations. In a similar effort, developed the Acute Pain in Neonates (APN-db) database, which utilizes facial expression and eye movement analysis to support early and objective pain detection in neonates, a group incapable of self-reporting. Their work highlights the value of non-invasive and data-driven methods for pain evaluation in non-verbal patients.

Expanding on these approaches, Husaini et al. [14] applied convolutional neural networks (CNNs) to classify pain intensity using facial data from the UNBC McMaster Shoulder Pain Archive. Their findings demonstrate the effectiveness of deep learning techniques in recognizing pain-related expressions across diverse clinical populations, supporting the broader applicability of automated pain detection systems. Collectively, these studies represent significant progress in the development of inclusive and personalized methodologies for pain assessment, particularly for vulnerable groups. By addressing the limitations of traditional tools, they contribute to more equitable and accurate clinical decision-making.

### 2.5. Facial Emotion Representation and Multi-Task Learning for Pain Assessment

Recent advancements in pain emotion detection have increasingly emphasized the integration of multi-task learning and enhanced feature representation to improve the accuracy and real-time applicability of automated systems. Nadeeshani, M. et al. [15], deep learning models that utilize facial AUs to predict emotional states associated with pain were developed. By leveraging AU-based analysis, this approach refined the detection of pain-related expressions, underscoring the critical role of facial muscle movement patterns in the accurate assessment of discomfort. This work laid a foundation for the integration of more nuanced facial analysis into deep learning frameworks.

Building upon this progress, ref. [16] introduced multi-order networks (MONET), a multi-task learning architecture that enhances AU detection by organizing tasks hierarchically. MONET’s design enables a more precise recognition of facial expressions linked to pain, contributing to the development of robust and scalable pain assessment systems. Extending the applicability of these techniques to critical care environments, Nerella et al. [17] introduced the Pain-ICU dataset, specifically curated to identify AUs indicative of pain in critically ill patients. Their study demonstrated that deep learning models trained on this dataset significantly outperformed conventional assessment tools, highlighting the clinical potential of AI-driven pain monitoring for real-time decision-making. Collectively, these studies illustrate the growing influence of multi-task learning and advanced feature representation in advancing the precision, efficiency, and adaptability of pain emotion detection across a range of medical contexts.

In parallel with multi-task learning, recent studies in micro-expression recognition and invariant image analysis offer promising directions for enhancing the robustness of pain detection frameworks. Zhou et al. [18] proposed a super-resolution approach for improving micro-expression recognition in low-resolution images, which could be beneficial in clinical scenarios where image quality varies. Additionally, Zhang et al. [19] and Mo et al. [20] introduced structural and rotation-invariant feature descriptors that improve facial analysis under pose variations and noisy inputs. Although these methods have not yet been widely applied to pain estimation, they present valuable opportunities for future integration to address challenges such as facial subtlety, head movement, and image consistency across clinical datasets.

### 2.6. Illumination and Environmental Factors in Pain Expression Recognition

Lighting variation remains a critical challenge in the accurate recognition of pain through facial expressions, as fluctuations in illumination can distort facial features and significantly impact the performance of computational models. To address this issue, ref. [21] proposed an advanced illumination normalization method that leverages multi-stage feature maps to enhance the analysis of facial color expressions. By systematically compensating for inconsistencies in lighting, this approach improves the reliability and robustness of pain detection systems, particularly under varying environmental conditions. This advancement is instrumental in maintaining model accuracy in real-world applications, where lighting cannot be consistently controlled.

The integration of illumination correction techniques into facial expression-based pain detection frameworks strengthens the overall effectiveness and generalizability of these systems. The existing research in this domain encompasses a wide array of methodologies, including deep learning architectures, neural networks, facial action unit analysis, and approaches tailored to special populations such as neonates and individuals with neurological impairments. By incorporating solutions to real-world challenges such as lighting variation, these studies contribute to the development of more precise, adaptable, and clinically viable pain assessment tools. Collectively, these innovations support the creation of accessible and scalable technologies capable of enhancing patient care across both clinical and non-clinical environments.

## 3. Methodology

This study aimed to improve the accuracy of pain level estimation by integrating facial expression recognition (FER) with a weighted analysis of specific AUs. As a preliminary step, facial images were subjected to normalization and scaling using standard image processing techniques to ensure consistency across samples and enhance analytical precision. Following this, the MediaPipe Face Mesh framework was utilized to detect up to 468 facial landmarks, which served as critical reference points for accurately identifying AU zones associated with pain-related expressions.

The selection of AUs—specifically, AU1 (inner brow raiser), AU6 (cheek raiser), AU9 (nose wrinkle), AU15 (lip corner depressor), AU17 (chin raiser), and AU44 (squint)—is grounded in prior empirical research that has demonstrated their strong association with pain expression. These AUs have been consistently validated as reliable indicators of pain intensity across a range of facial conditions and states.

The proposed pain level estimation framework incorporates a weighted contextual representation derived from FER in conjunction with the calculated intensities of the selected AUs. This integrated methodology enables the capture of subtle variations in facial dynamics, thereby significantly enhancing the precision and robustness of pain assessment. A comprehensive overview of the methodological workflow and its component processes is illustrated in Figure 1, providing a clear depiction of the analytical framework employed in this study.

### 3.1. Normalization

In the initial stage of the proposed methodology, the input image frames are subjected to facial scaling and normalization procedures to ensure consistency and enhance the accuracy of subsequent analytical processes. The facial images utilized in this study were obtained from a publicly available dataset hosted on Kaggle.com. The use of a standardized and accessible dataset enables a systematic analysis of facial expressions in relation to pain assessment. The dataset comprised a total of 4084 images, each categorized according to distinct facial expressions. A detailed breakdown of these categories is presented in Table 1.

To ensure consistency and reliability in pain level estimation, the initial preprocessing stage involves scaling the detected face to a standardized size. This resizing process is essential for maintaining uniformity across varying input frames, which may differ in size and resolution. Following this, normalization is performed by adjusting pixel intensity values to a common scale, typically within the range of 0 to 1, in order to minimize the influence of lighting variations that could adversely affect the detection of pain-related facial features. Additionally, face alignment is conducted by identifying the key facial landmarks, such as the eyes and the nose, and rotating the facial region to achieve consistent orientation across all frames. These preprocessing steps collectively establish a robust foundation for accurately identifying pain-relevant facial features and quantifying pain intensity in subsequent analytical stages.

The implementation of this methodology was carried out using a Python version 3.9.7, employing the Python Imaging Library (PIL) and Matplotlib version 3.4.3. The script processes facial images associated with varying levels of pain and discomfort to ensure standardization for subsequent analysis. Each image is resized to a consistent width while preserving its original aspect ratio, facilitating uniform feature extraction. To further enhance image quality, the Unsharp Mask filter is applied, emphasizing edges and fine details to sharpen facial features, such as wrinkles, muscle tension, and micro-expressions, indicative of pain. Moreover, adjustments in contrast and brightness are employed to enhance the visibility of subtle facial cues, ensuring that critical expressions, such as furrowed brows, tightened lips, and squinting eyes, are consistently and accurately detected. This comprehensive preprocessing pipeline enhances the precision and objectivity of pain level estimation throughout the study.

### 3.2. Face Mesh Detection

After image normalization, facial landmarks are detected using MediaPipe’s Face Mesh technology, which provides up to 468 distinct facial points. These landmarks cover the key facial regions, such as the eyes, eyebrows, nose, lips, and jawline, and serve as the basis for estimating action units (AUs) relevant to pain detection.

The precision of MediaPipe’s landmark detection enhances the spatial resolution and consistency of AU localization, especially for subtle muscle movements commonly associated with pain expressions. Each facial zone defined by the mesh corresponds to specific AUs (e.g., AU6 around the eyes, AU15 at the corners of the mouth), allowing for fine-grained mapping between landmark clusters and muscular activations.

Compared to traditional landmarking approaches such as the supervised descent method (SDM), MediaPipe offers improved automation, speed, and robustness under varied lighting and pose conditions. This not only increases detection accuracy, but also reduces the dependency on manual annotation.

By integrating MediaPipe’s Face Mesh with AU-based analysis, the system achieves a higher level of reliability and context-aware interpretation of facial expressions for pain level estimation. A visual representation of these analytical steps, along with the AU-mapped zones, is provided in the following illustration.

### 3.3. AU Zone Calculation

As illustrated in the accompanying diagram, this stage of the methodology focuses on the detection of specific facial feature points that are critical for subsequent analytical processes. These feature points are systematically identified by defining zones of interest across the face, each corresponding to regions associated with the key facial AUs. This zoning approach ensures targeted analysis of areas most relevant to the assessment of pain-related facial expressions.

AUs are standardized representations of the individual facial muscle movements widely utilized in facial expression analysis. Each AU corresponds to a specific and quantifiable muscular action, such as the raising of an inner brow (AU1), the tightening of lips (AU23), or other subtle movements, that collectively convey a range of emotional and physiological states. The identification and quantification of AUs serve as the foundation for analyzing facial dynamics, offering critical insights into non-verbal indicators such as pain, discomfort, and distress. This framework is particularly vital for automated pain detection systems and expression-based behavioral analysis, as it facilitates the interpretation of nuanced facial variations that might otherwise go unnoticed.

To ensure accurate detection of these subtle muscular activities, the system incorporates advanced algorithms and machine learning models trained specifically to recognize the distinctive patterns associated with each AU. As illustrated in Figure 2, these models are capable of detecting even the most minor deviations in facial expressions, thereby enabling a more granular and reliable analysis of pain-related cues.

Facial features and their corresponding characteristics serve as the foundation for identifying specific regions in which AUs are analyzed for detailed pain assessment. The AUs utilized in this study—namely AU1, AU6, AU9, AU15, AU17, and AU44—are linked to the key muscle movements that reflect emotional expressions, physiological responses, and pain-related facial dynamics. The associated diagram highlights the critical facial zones, including the forehead, eyes, nose, and mouth, where even subtle muscular changes may signal varying degrees of discomfort or distress.

By concentrating on these anatomically and psychologically significant regions, the system is capable of detecting micro-expressions and involuntary muscle activity, allowing for a fine-grained interpretation of facial responses to pain. For instance, AU1, located in the forehead region and involving the raising of the inner brows, is traditionally associated with emotions such as surprise or curiosity, but may also indicate acute discomfort when viewed in the context of pain. Similarly, AUs situated around the eyes, such as AU6 (cheek raiser) and AU44 (squint), are instrumental in detecting squinting or eye tightening, both of which are frequently observed in individuals experiencing heightened pain. Mouth-related AUs, including AU9 (nose wrinkle), AU15 (lip corner depressor), and AU17 (chin raiser), capture expressions such as lip tightening, grimacing, and stretching of the mouth, which are characteristic of moderate to severe pain states.

The structured grid presented in Figure 3 provides a systematic framework for mapping and measuring these facial changes, thereby ensuring a high level of precision in pain detection, emotion recognition, and clinical diagnostics. By quantifying the intensity and frequency of AU activations, the system facilitates an objective and data-driven approach to pain assessment. To further elaborate on these relationships, Table 2 presents a comprehensive classification of the AU regions, their anatomical locations, and the corresponding facial expressions observed at various levels of pain intensity. This structured mapping supports the development of scientifically grounded, automated systems for accurate and reliable pain detection.

Each of the selected AUs corresponds to facial muscle activations strongly linked to pain-related expressions. For instance, AU4 (brow lowerer) and AU15 (lip corner depressor) are commonly activated during pain and emotional distress, while AU6 and AU9 show modulation depending on the emotional context (e.g., joy vs. discomfort). These AUs are used to calculate both the baseline pain score (Equation (3)) and the emotion-weighted PLE score (Equations (4) and (5)). Therefore, the same AU measurements contribute to both pain detection and emotion modulation processes.

Through this process, the framework ensures that all relevant facial feature points and associated AUs are accurately captured and analyzed, enabling a comprehensive interpretation of the subject’s facial expressions and the corresponding pain levels. Following the detection of facial landmarks, the methodology proceeds along two parallel analytical paths: facial expression recognition and pain scale estimation. As illustrated in the accompanying diagram, this bifurcated approach facilitates both the classification of general emotional expressions and the quantification of pain intensity, thereby enhancing the robustness and precision of the overall pain assessment system.

The models employed in this system are specifically designed to detect subtle facial changes indicative of pain, including micro-expressions and involuntary muscle movements. By analyzing the defined AUs, the system facilitates a detailed assessment of facial dynamics, enabling precise and objective evaluation of pain intensity. The zones of interest and the corresponding AUs utilized in this framework are grounded in well-established psychological and anatomical research, ensuring that the detected facial movements are reliably correlated with pain-related expressions. For example, AU15, commonly associated with smiling, may also reflect pain-masked expressions when observed in conjunction with other AUs. By capturing such nuanced interactions, the system extends beyond subjective visual observation, offering a data-driven approach to pain assessment.

To further enhance accuracy, six key facial regions have been identified and mapped to specific AUs, as illustrated in Figure 3. These regions are strategically selected based on their relevance to facial muscle activity associated with pain. Through this mapping, the system is capable of detecting even the subtlest changes in muscle activation, thereby improving the sensitivity and reliability of discomfort or pain detection across a range of facial expressions.

To accurately measure the intensity of facial muscle movements, the displacement between landmark points is calculated using the Euclidean distance equation:(1)$$D_{{AU}_{i}}=\sqrt{{\left(x_{2}+x_{1}\right)}^{2}+{\left(y_{2}+y_{1}\right)}^{2}}$$
where $D_{{AU}_{i}}$ is the displacement representing the i^th^ action unit’s movement, $\left(x_{1},y_{1}\right)$ is the standard (neutral) landmark point, and $\left(x_{2},y_{2}\right)$ is the activated landmark point during the expression.

These distance values reflect the degree of muscle activation within each AU zone. Once computed, they are normalized and mapped as AU intensity values, forming the basis for further analysis in the next stages.

These AU intensity values are then integrated into various analytical equations. In Equation (2), the weighted sum of AUs is used for facial expression recognition (FER). In Equation (3), a pain intensity score is estimated from the specific AUs known to correlate with discomfort, while Equation (4) uses the maximum AU intensity from each zone to assess pain more precisely. Finally, Equation (5) combines both AU-based pain estimation and expression-weighted adjustments to produce a comprehensive pain level score.

This distance-to-AU-to-score process ensures a data-driven approach to objectively quantify facial movements and accurately interpret pain intensity.

### 3.4. Emotion Recognition and Scale

In this analytical path, the detected facial feature points are processed to recognize facial expressions. A pre-trained model is employed to classify these expressions based on the spatial configuration and dynamic movement of the identified feature points. The classification system categorizes expressions into fundamental emotional states, including happiness, sadness, anger, surprise, fear, and disgust. Recognizing these emotional cues offers valuable insights into the subject’s affective state, which is essential for contextualizing pain-related expressions. The integration of emotion recognition not only supports a more holistic interpretation of facial behavior, but also aids in distinguishing between pain-induced and emotion-driven facial changes. The general formulation for facial expression recognition (FER) can be represented as follows:(2)$$FER=\sum_{i=1}^{k}W_{i}AU_{i}$$
where $W_{i}$ is the weight assigned to the i^th^ AU, ${AU}_{i}$ is the detected intensity of the i^th^ AU, and k is the total number of AUs considered for the specific facial expression. This equation aggregates the weighted intensities of multiple AUs to determine the overall emotional expression. Each AU corresponds to a specific facial muscle movement, and the weights $W_{i}$ are determined based on the importance of each AU in conveying particular emotions. The sum of these weighted values gives a comprehensive measure of the detected emotion, allowing the system to accurately classify and interpret facial expressions.

This process is particularly critical in applications such as pain assessment, where the ability to discern subtle variations in facial expressions can significantly enhance the accuracy of evaluating an individual’s emotional and physical state. By incorporating the weighted contributions of various AUs, the facial expression recognition (FER) system improves its capacity to distinguish between expressions that may appear visually similar yet stem from different underlying causes. For instance, the system is capable of differentiating between a grimace indicative of pain and a frown associated with anger—two expressions that, while superficially alike, reflect fundamentally distinct emotional or physiological experiences. This level of granularity contributes to more precise and context-aware interpretations within automated pain detection frameworks.

To quantitatively assess the relationship between emotional expressions and pain perception, each emotion is assigned a constant value representing its relative intensity and potential correlation with pain. The proposed emotion–pain correlation scale is as follows: happiness, 0.8; surprise, 0.9; neutrality, 1.0; disgust, 1.2; fear, 1.4; anger, 1.6; sadness, 1.7.

This scale serves as a standardized framework for quantifying facial expressions, thereby enabling a more objective interpretation of emotional states in the context of pain detection. The assigned values reflect both the intensity of each expression and its associated impact on facial muscle activity, particularly as it pertains to pain-related dynamics. Emotions such as sadness, fear, and anger are frequently observed in conjunction with higher levels of pain, whereas such emotions as happiness and neutrality tend to align with lower pain intensities. By incorporating these predefined weights into pain assessment models, the system enhances its capacity to distinguish between psychological distress and physical discomfort, thereby supporting more accurate clinical evaluations.

Advancements in automated facial AU detection have significantly contributed to the development of pain level estimation (PLE), particularly through the use of the Prkachin and Solomon pain intensity (PSPI) metric. PLE is derived from the detection and analysis of the key AUs, allowing for a quantifiable and reproducible evaluation of pain intensity. By accurately identifying facial muscle movements associated with specific AUs, researchers can focus on the facial regions most indicative of pain, thereby improving detection precision.

To increase the reliability and validity of AU-based pain assessment, six primary AUs—selected from Table 2—were analyzed in detail. Additional reference points were integrated to refine the evaluation process, ensuring enhanced sensitivity to subtle facial cues. This targeted approach reinforced the link between specific muscle activations and varying levels of pain, contributing to a more standardized and replicable assessment framework. The system employed the PSPI scale to score the detected AUs, combining these values to generate an expression pain scale. This output served as an objective and consistent measure of pain intensity, supporting both real-time and retrospective analyses.

By integrating emotional expression weighting with AU-based pain evaluation, the proposed framework offers a comprehensive and accurate method for automated pain detection.

Ultimately, the integration of facial expression recognition (FER), emotion–pain correlation scaling, and AU-based analysis converges into the final pain level estimation model. The weighted contributions of each action unit, modulated by the emotional context detected through FER, are combined in Equation (5) to produce the final ${PLE}_{w}$ score. This score reflects a more comprehensive understanding of pain by considering both the physiological (AU intensity) and psychological (emotional state) components of facial expressions. The use of this hybrid approach ensures that the system can accurately estimate pain even when expressions are influenced by emotional masking, overlap, or ambiguity—making it more robust for real-world clinical applications.

### 3.5. Facial Scale for Pain Assessment

Following the previous steps, the results from facial expression recognition and the expression pain scale are weighted and combined to assess the overall pain level. This integration is represented in the diagram by the summation symbol (Σ), indicating the combination of data from both analysis paths.

In this step, the system takes the recognized facial expressions and the quantified pain intensity scores, applying appropriate weights to each factor. This weighting process ensures that both the emotional context and the specific pain-related facial AUs are accurately represented in the final assessment.

The combined data are then used to create a comprehensive facial scale for pain assessment. This scale integrates the emotional and pain-specific indicators to produce a single, unified pain level score. The pain level score is a quantitative measure that reflects the subject’s overall pain intensity based on their facial expressions. We were able to categorize the levels of pain according to each pain score range as follows:No pain (painless);Pain level 1: mild (indicating minimal discomfort or irritation);Pain level 2: moderate (suggesting a noticeable but tolerable level of discomfort);Pain level 3: severe (representing a significant and distressing level of pain);Pain level 4: very severe (signifying intense and incapacitating pain);Pain level 5: worst pain (indicating the highest possible level of pain and extreme distress).

The PSPI scale quantifies facial expressions of pain using six core action units (AUs): AU4 (brow lowerer), AU6 (cheek raiser), AU7 (lid tightener), AU9 (nose wrinkler) or AU10 (upper lip raiser), and AU43 (eye closure). Each AU is scored on an ordinal scale from 0 to 5 based on its intensity, except AU43, which is scored as 0 or 1. The maximum PSPI score is 15, representing the sum of these AU intensities; 0 stands for no facial indication of pain, 15—for a strong facial expression of pain.

The Prkachin and Solomon pain intensity (PSPI) scale quantifies facial expressions of pain using six core action units (AUs): AU4 (brow lowerer), AU6 (cheek raiser), AU7 (lid tightener), AU9 (nose wrinkler) or AU10 (upper lip raiser), and AU43 (eye closure). Each AU is scored from 0 to 5 based on its intensity, except AU43, which is binary (0 or 1). This results in the total PSPI score ranging from 0 (no facial indication of pain) to 15 (strong pain expression), making it a widely used benchmark in automated pain recognition.

In this study, the PSPI metric was adopted as a baseline framework for computing the initial pain level estimation (PLE) score. The selected AUs—AU4, AU6 or AU7, AU9 or AU10, and AU43—represent specific muscle activations that are frequently observed during pain. Equation (3) formulates this baseline score as follows:PLE = AU4 + max(AU6|AU7) + max(AU9|AU10) + AU43(3)

This baseline score is subsequently refined through emotion-based weighting, as described in Equations (4) and (5). The use of the PSPI ensures compatibility with prior studies and provides a robust foundation for evaluating the effectiveness of the proposed weighted pain estimation framework.

AU4 contributes to pain estimation by indicating eyebrow contraction, which is commonly associated with frowning and discomfort. The equation takes the maximum value between AU6 (cheek raiser) and AU7 (lid tightener), ensuring that the most pronounced muscle movement linked to pain is accounted for. Similarly, the maximum value between AU9 (nose wrinkler) and AU10 (upper lip raiser) captures nasal and upper lip contractions, both of which are key indicators of pain expressions. Finally, AU43 (eye closure) reflects the instinctive response of shutting the eyes tightly due to discomfort or intense pain.

By combining these essential facial features, the PLE equation effectively captures a broad spectrum of pain-related facial expressions. The use of the maximum function ensures that the strongest pain indicator in each category contributes to the final estimation.

The researcher-specified pain intensity Equation is as follows:PLE = max(AU1) + max(AU6) + max(AU9) + max(AU15) + max(AU17) + max(AU44)(4)

The selection of AU zones on the face from Figure 3 and the application of the pain level estimation Equation (3) allowed us to determine the most significant AU activations for pain intensity estimation. Each AU zone consists of multiple facial landmarks that capture different levels of muscle activation. Since facial movements vary in intensity across different regions, the maximum value from each AU zone is selected to represent the most significant activation in that area. This ensures that the equation effectively captures the strongest pain-related facial expressions.

For instance, AU1 (inner brow raiser) is measured in the forehead region, where multiple points track eyebrow elevation. The highest detected intensity from these points is selected as max(AU1). Similarly, AU6 (cheek raiser), which indicates cheek muscle contractions around the eyes, is assigned its strongest detected value as max(AU6). AU9 (nose wrinkler), responsible for nose wrinkling, is measured across the nasal area, and its highest activation is used as max(AU9). In the lower face region, AU15 (lip corner depressor), which reflects the downward movement of the lips, contributes with its strongest detected contraction as max(AU15). AU17 (chin raiser), which measures chin muscle tension, is represented by its maximum activation as max(AU17). Lastly, AU44 (squint), associated with involuntary eye tightening due to discomfort, is included in the equation by selecting its highest intensity as max(AU44).

After determining the maximum AU values from each region, the final pain level estimation (PLE) is computed by summing these values. This method ensures that the most prominent pain-related facial activations are accounted for, making the estimation more accurate and sensitive to subtle changes in facial expressions by focusing on the strongest AU activations from different facial zones.

The weighted pain level estimation Equation (5) integrates both facial expression analysis and pain level estimation (PLE) from AUs to calculate the overall pain level. This approach ensures that both facial muscle activation and contextual variations in facial expressions are considered, providing a more comprehensive and refined assessment of pain perception. By incorporating weighted adjustments, the model improves the accuracy of pain detection, distinguishing between facial movements caused by pain and those influenced by other factors. The equation is as follows:(5)$${PLE}_{W}^{z[1–6]}=\sum_{W=1}^{m}P_{s}FE_{W}$$
where ${PLE}_{W}^{z[1–6]}$ is the weighted values of seven expression categories—represented by m = 7—used to compute ${PLE}_{w}$; $P_{s}$ is Equation (4), yields the PLE pain level; and ${FE}_{w}$ represents the values derived from facial emotion detection [happiness, sadness, fear, disgust, anger, surprise, neutrality].

To enhance the reliability of pain level estimation, a weighted analytical mean is employed in this study. This approach allows each AU to contribute to the final pain score proportionally to its relevance under a given emotional context.

Equation (5) represents the final weighted pain level, which integrates AU-based pain estimation with an expression-based adjustment mechanism. The term $P_{s}$ denotes the pain intensity score derived from facial AU activations, while ${FE}_{w}$ accounts for the influence of various facial expressions on pain perception. By incorporating expression-based weighting, this approach enhances the accuracy of pain assessment, addressing the challenges posed by variations in facial dynamics that may affect traditional AU-based pain detection methods.

This hybrid approach strengthens pain recognition by combining objective biological pain markers (AUs) with additional expression-based factors, making the model more adaptable to diverse real-world conditions. The weighting mechanism ensures that expressions commonly linked to distress, such as fear and sadness, contribute more significantly to pain estimation, whereas neutral or positive expressions may lower the perceived intensity. This enables the model to differentiate between pain-driven facial movements and those influenced by other expressions, improving the overall assessment reliability.

### 3.6. Algorithmic Framework

Algorithm 1 defines the foundational process for computing the raw pain score Ps, derived by aggregating the maximum activations of the selected action units (AUs) known to be associated with pain expression. This raw score reflects the intensity of facial muscle contractions related to pain without any influence from contextual or emotional factors. However, facial expressions are often influenced not only by physiological pain, but also by emotional states such as fear, sadness, or anger, which may alter the perceived intensity of facial signals. To address this, Algorithm 2 introduces a weighting mechanism that modulates the raw pain score based on the individual’s emotional expression category. By incorporating a predefined weight corresponding to each emotional state, this adjustment allows the model to account for contextual variations and better distinguish between facial activations caused by genuine pain and those influenced by concurrent emotional expressions. The result is a weighted pain estimate, denoted as ${PLE}_{W}^{z[1–6]}$, that enhances the model’s robustness and interpretability in real-world applications.
**Algorithm 1** Compute the pain score from facial action units **Procedure** ComputePainScore(AU1, AU6, AU9, AU15, AU17, AU44)        PLE ← max(AU1) + max(AU6) + max(AU9) + max(AU15) + max(AU17) + max(AU44)        $P_{s}$ ← PLE        **return** $P_{s}$ **end procedure**

**Algorithm 2** Emotion-weighted pain computation
**Input**:- emotion_label ∈${FE}_{w}$-$\;{FE}_{w}$ = [happiness, sadness, fear, disgust, anger, surprise, neutrality]-$\;\mathrm{emotion}\_\mathrm{weights}:\;a\;\mathrm{mapping}\;\mathrm{from}\;{FE}_{w}$ → numeric weights-$\;\mathrm{pain}\_\mathrm{score}:\;\mathrm{the}\;\mathrm{raw}\;\mathrm{pain}\;\mathrm{score}\;(P_{s}$), computed from facial landmarks (Algorithm 1)- PLE: pain level estimate (equal to pain_score in this context)**Procedure** WeightedPainComputation(pain_score, PLE, emotion_label, emotion_weights)    **if** pain_score ≠ None and PLE ≠ None then          
**if** emotion_label ∈${FE}_{w}$ then                  pain_weighted ← pain_score × emotion_weights[emotion_label]                  ${PLE}_{W}^{z[1–6]}$ ← PLE × emotion_weights[emotion_label]                  $\mathrm{return}\;(\mathrm{pain}\_\mathrm{weighted},\;{PLE}_{W}^{z[1–6]}$)**          end** if**    end** if**    return** none
**end procedure**



This two-stage algorithmic framework distinctly separates the biological foundation of facial pain signaling—captured through action unit (AU) activations—from the contextual modulation introduced by emotional expressions. The modular design allows for flexible integration of emotional influence, thereby enhancing the system’s ability to differentiate between pain-induced facial movements and those arising from emotional states. This separation improves both the interpretability and reliability of automated pain assessment models in complex, real-world scenarios.

Pain level determination within this framework is achieved through a weighted evaluation of facial expressions, wherein computational adjustments are applied using the proposed algorithms as part of the experimental methodology. This process ensures a more standardized and accurate interpretation of pain intensity, ultimately supporting the advancement of robust, automated pain detection systems for clinical diagnostics and behavioral research.

## 4. Experiments

In this study, a dataset comprising 4084 annotated facial images from 25 subjects was utilized. These images encompassed a diverse range of pain levels and emotional expressions and were manually labeled for action units (AUs) and pain intensities. While the dataset size was moderate compared to large-scale vision benchmarks, its annotation quality and expression diversity made it suitable for experimental validation in pain estimation studies.

To ensure generalizability and reduce overfitting, data augmentation techniques, such as rotation, flipping, and scaling, were applied. Additionally, the dataset includes varied facial orientations, lighting conditions, and demographic diversity to reflect real-world conditions. Future work will focus on expanding the dataset to enhance statistical significance and model robustness across broader populations.

### 4.1. Normalization

A crucial step in this analysis is the normalization of facial images, which ensures consistency and accuracy in subsequent evaluations. Normalization involves adjusting the lighting, contrast, and size of images to make them suitable for precise feature detection. Figure 4 below illustrates this process by displaying both the original and normalized versions of the selected facial images. Normalization enhances the clarity and comparability of facial features, thereby improving the accuracy of subsequent pain level assessments. This step is fundamental in establishing a reliable framework for analyzing facial expressions in the context of pain detection.

Normalization involves standardizing lighting, contrast, and facial alignment, which helps reduce variability across images. This process enhances the consistency of facial features, allowing the system to accurately detect subtle muscle movements and micro-expressions associated with pain levels. By minimizing external influences such as uneven lighting or pose variations, normalization improves the reliability and precision of automated facial analysis, ensuring a more accurate assessment of pain-related facial dynamics.

### 4.2. Face Mesh Detection

After normalizing the images, MediaPipe’s Face Mesh (version 0.10.3) was utilized to detect up to 468 facial landmarks, enabling precise identification of the AUs crucial for pain detection. The MediaPipe’s Face Mesh enhances the accuracy of identifying detailed facial structures such as the eyes, mouth, and nose, thereby improving AU analysis for pain level assessment. The detailed steps involved in the Face Mesh detection process are as follows:The normalized facial images are input into the MediaPipe’s Face Mesh module;The module identifies significant facial landmarks required for subsequent action unit (AU) computation;The detected landmark coordinates are recorded in a three-dimensional format (x, y, z) to facilitate subsequent analytical stages.

Figure 2 illustrates detailed mapping of facial landmarks.

### 4.3. Au Zone Calculations

In Figure 5, the facial landmark detection process using MediaPipe is presented as a foundation for analyzing fine-grained facial muscle movements. The left panel displays a structured mesh overlay that represents the three-dimensional facial geometry derived from the detected landmarks. In contrast, the right panel emphasizes representative points within the predefined action unit (AU) zones, which target anatomically significant regions such as the eyebrows, eyes, nose, mouth, and chin. These zones are selected based on their relevance to muscle activation patterns associated with pain expression. By focusing on these localized regions, the proposed system aims to enhance the accuracy and objectivity of automated pain intensity assessment through dynamic facial analysis.

After determining the facial landmarks, we proceeded to define the zones for the AUs as follows:Inner brow raiser (AU1): located between the eyebrows, represented by the color green;Cheek raiser (AU6): located behind the eyes, shown in blue;Nasal wrinkle (AU9): indicated by the color black;Lip corner depressor (AU15): situated close to the corners of the mouth, represented by the color red;Chin raiser (AU17): located near the chin creases, shown in cyan/aqua;Squint (AU44): covers the surrounding regions, represented by the color yellow.

These zones can be seen in Figure 5.

### 4.4. Emotion Recognition and Scale

In this study, facial images from a dataset were utilized to analyze and predict facial expressions through Equation (2), which processes the images to generate the corresponding numerical values. This method converts facial features into quantifiable data, enabling precise expression prediction and enhancing the accuracy of the analysis.

The dataset, designed to identify seven distinct facial expressions, was rigorously evaluated using a widely recognized method to ensure reliability. Additionally, the system provided numerical probability scores for each expression, allowing for a detailed assessment of the likelihood of different facial expressions. The results, which illustrate the predicted probabilities for each category, are summarized in Table 3.

In Figure 6, the confusion matrix for expression prediction evaluates the model’s performance in classifying facial expressions by comparing actual labels to the predicted outcomes. The diagonal cells represent correct classifications, where the model accurately identified an expression, while the off-diagonal cells indicate misclassifications, highlighting instances where the model incorrectly assigned an expression label. The color intensity reflects the frequency of occurrences, with darker shades representing higher accuracy in classification and lighter shades indicating fewer instances.

This confusion matrix provides valuable insights into the model’s strengths and weaknesses in distinguishing between different facial expressions. A higher concentration of values along the diagonal suggests a strong classification performance, whereas significant misclassifications in off-diagonal cells may indicate challenges in differentiating certain expressions. By analyzing these patterns, researchers can identify areas for improvement, such as refining feature extraction, adjusting classification thresholds, or enhancing dataset diversity to improve the robustness of expression recognition models.

In Table 4, F1-score reflects the balance between precision and recall, indicating the overall classification quality. AUC measures the model’s ability to distinguish between expressions, while mAP summarizes the precision–recall performance across all classes, highlighting effectiveness in multi-class recognition.

### 4.5. PSPI and Pain Scale

The analysis of pain intensity is refined by weighting the results of facial expression recognition. Equation (4) calculates pain levels from specific locations within each AU, ensuring precise quantification. Then, Equation (5) identifies the AU with the highest value in each facial zone, highlighting the key pain indicators. This weighting process enhances the accuracy of computed pain intensity, aligning it closely with observed emotional expressions. As shown in the accompanying table, this method significantly improves the precision of pain measurements, offering a more accurate reflection of the subject’s true emotional state. From Figure 7, after normalizing the images and assigning AU zones, the pain level is calculated using Equation (4), based on distance values from each AU point.

All facial images shown in this figure were used with informed consent and anonymized to protect the identity of participants. The study was conducted in accordance with ethical research standards and institutional guidelines for human subject research.

Example images of changes in facial muscle points illustrate the variations in facial landmarks that occur due to muscle movements associated with pain expressions. These images show tracked facial points in both neutral and pain conditions, particularly within the key AU zones. The distance between a neutral landmark (green) and a pain landmark (blue) represents the degree of muscle displacement. To quantify this, facial landmarks are extracted from both expressions using a tracking algorithm such as MediaPipe’s Face Mesh. Each corresponding landmark pair is analyzed to determine the extent of spatial displacement, which reflects underlying muscle activity. Larger displacements in areas such as the eyebrows, eyes, and mouth typically indicate higher levels of perceived pain. This distance-based measurement enables objective and interpretable pain assessment based on facial changes.

In Table 5, Equation (3), a commonly used pain intensity equation (PLE and ${PLE}_{w}$), PLE represents the percentage accuracy obtained using a standard pain assessment equation. It is a common metric used to evaluate pain intensity without any weighting factors. ${PLE}_{w}$ is the weighted version of the PLE equation, incorporating different weights for factors influencing pain assessment to enhance accuracy.

Equation (4), the researcher-specified pain intensity equation (${PLE}^{z[1–6]}$), is specified by researchers and tailored to consider various specified factors (z[1–6]) that influence pain intensity. It aims to provide a more customized and precise pain assessment compared to the common equation.

Equation (5), overall pain level using a weighted combination (${PLE}_{W}^{z[1–6]}$), represents the overall pain level calculated using a weighted combination of various influencing factors. By integrating weights into the assessment, this method aims to offer the most accurate reflection of pain intensity, combining both common and researcher-specified factors.

## 5. Discussion

This study explored the intricate relationship between facial expressions and pain perception by analyzing facial features to quantify pain levels. Since facial expressions serve as both a natural response and an indicator of physical discomfort, accurately distinguishing between these aspects is essential for improving pain assessment methodologies. By examining variations in facial muscle dynamics, the proposed approach integrates both physiological responses and contextual facial cues to enhance pain level estimation. Unlike traditional pain detection methods that rely solely on AUs or predefined pain scales, this study introduces a weighted pain assessment model that adjusts the pain intensity score based on facial cues. To refine accuracy, a weighting mechanism is applied in later stages, allowing the system to differentiate between pain-related and non-pain-related facial expressions more effectively. Although variations in facial region identification pose certain challenges, the results demonstrate that this hybrid approach significantly improves accuracy compared to previous models, reinforcing its reliability in pain assessment.

Facial expressions play a pivotal role in understanding both physical and psychological pain, but their overlap can sometimes create ambiguity. To address this, researchers developed the pain level expression (PLE) model, which leverages the key facial regions associated with pain-related responses to create a more refined pain estimation method. This model was validated using a dataset of 4084 facial images from 25 individuals, ensuring a diverse and comprehensive analysis. Unlike conventional pain measurement systems that rely solely on AU activations, this approach integrates a weighting mechanism to dynamically adjust pain intensity predictions. For instance, negative expressions such as fear, sadness, or disgust are often correlated with higher pain perceptions, while neutral or relaxed expressions might indicate lower pain levels. By incorporating these modulation techniques, the model achieves a more realistic and nuanced interpretation of pain-related facial dynamics.

The findings of this research address the existing limitations related to accuracy challenges in automated pain detection systems, especially when interpreting ambiguous or subtle facial expressions influenced by contextual variables. By introducing a weighted mechanism that integrates emotional context into AU-based analysis, the proposed model provides a more nuanced and precise interpretation of pain-related facial expressions. Consequently, this advancement offers significant theoretical implications by expanding current knowledge regarding the relationships between emotional facial expressions and pain perception. Practically, the developed method can significantly enhance clinical practices, facilitating improved diagnostic accuracy and timely interventions for patients with communication impairments or those unable to verbally express their pain, thus promoting better patient outcomes in healthcare settings.

While this study focused on a core set of six AUs, other units such as AU4 (brow lowerer) and AU10 (upper lip raiser) are also known to co-occur with pain expressions and may be integrated into future model extensions to improve granularity and emotional context sensitivity.

The PLE model’s weighted framework ensures that pain assessment is not only based on direct AU activations, but also considers expression-based influences, making it more adaptive to real-world applications. This methodology ultimately supports precise and reliable pain detection, which is crucial for applications in clinical pain management, automated healthcare systems, and patient monitoring technologies. The ability to quantify pain in a data-driven manner significantly enhances pain assessment accuracy, providing a robust tool for medical professionals and researchers working towards better pain detection and management strategies.

Figure 8 presents a linear graph comparing the average pain levels across different emotional states, utilizing PLE (pain level estimation) and ${PLE}_{w}$ (weighted pain level estimation). The x-axis represents various emotions, while the y-axis categorizes pain levels ranging from no pain to level 5. The graph highlights the differences in pain perception when incorporating emotion-based weighting, demonstrating how certain emotional states influence the estimation of pain intensity.

The ${PLE}_{w}$ (red line) consistently registered higher pain levels than PLE (blue line), particularly for negative emotions such as disgust, fear, anger, and sadness. This suggests that the weighted model better reflects pain perception, as it accounts for the impact of emotional intensity on pain experiences. Notably, such emotions as happiness, surprise, and neutrality showed minimal differences between PLE and ${PLE}_{w}$, indicating that these emotions have a less pronounced effect on perceived pain. In contrast, the sharp increase in ${PLE}_{w}$ for sadness, anger, and fear suggests that strong negative emotions heighten pain perception, aligning with psychological research indicating that emotional distress can amplify physical discomfort.

This analysis reinforces the effectiveness of emotion-weighted pain assessment models, demonstrating how ${PLE}_{w}$ enhances traditional pain estimation methods by integrating both facial expression analysis and emotional influences. The ability of ${PLE}_{w}$ to differentiate pain levels across emotional states makes it a valuable tool for automated pain detection and clinical pain management. By providing a more nuanced and realistic pain assessment, this approach contributes to improving patient care, personalized treatment plans, and AI-driven healthcare applications.

Figure 9 illustrates the pain level estimation for individuals expressing happiness over a 60 s period, highlighting the relationship between positive facial expressions and pain perception. The graph compares PLE (blue) and ${PLE}_{w}$ (red), where PLE represents the standard pain estimation, while ${PLE}_{w}$ incorporates weighted adjustments based on expression-based influences. The fluctuations in ${PLE}_{w}$ values suggest that happiness contributes to a reduction in perceived pain levels, as the weighted pain estimation remained lower than the standard PLE.

Figure 10 presents the pain level estimation for individuals expressing negative emotions (sadness) over a 60-s interval. The graph compares PLE (blue) and ${PLE}_{w}$ (red), where PLE denotes the standard pain level estimation, and ${PLE}_{w}$ represents a weighted version that incorporates the influence of facial expressions. Unlike the previous scenario involving happiness, the ${PLE}_{w}$ values in this figure consistently exceed the unweighted PLE estimates. This indicates that the presence of negative expressions amplifies the perceived intensity of pain, resulting in higher estimations when emotional weighting is applied. Peaks in the ${PLE}_{w}$ curve often cross into higher pain level thresholds (e.g., Pain Level 4 and 5), highlighting the significant impact of emotional context on pain perception.

The comparison between these two expression-based conditions demonstrates that facial dynamics significantly influence pain estimation, supporting the use of weighted pain detection models. By integrating facial expression analysis and AU-based pain estimation, this approach enhances the accuracy and adaptability of pain monitoring systems. The findings suggest that incorporating contextual facial cues into automated pain assessment can lead to more precise and individualized pain management strategies, benefiting clinical applications and AI-driven healthcare solutions.

While the proposed linear model offers advantages in terms of simplicity, interpretability, and computational efficiency—especially in clinical applications where transparency and real-time execution are essential—it may not fully capture the complex nonlinear relationships between facial expressions and pain intensity. Recent studies employing attention-enhanced facial expression recognition (FER) architectures have demonstrated superior performance by dynamically weighting salient facial regions and contextual cues. Although our method does not currently benchmark against such architectures, we acknowledge this as a limitation. Future work will explore integrating attention mechanisms or transformer-based models to enhance spatial representation and improve robustness, particularly in scenarios involving subtle or ambiguous expressions.

## 6. Conclusions

This research introduces an innovative weighted analysis method for pain intensity estimation, combining facial expression recognition (FER) and action unit (AU) analysis. The proposed method enhances the accuracy and reliability of pain assessment, particularly in scenarios where verbal communication is not possible, such as in critically ill patients or special populations. By utilizing a dataset of 4084 images from 25 subjects, the method demonstrated superior performance compared to traditional approaches. The integration of facial expression analysis with AU-based assessment significantly improved the precision of pain level estimates, highlighting the importance of considering both facial cues and physiological responses in pain detection.

The proposed method successfully met the research objectives by significantly improving pain estimation accuracy from 83.37% to 92.72%. These results confirm the effectiveness of integrating facial expression weighting into AU-based pain analysis, demonstrating enhanced robustness and accuracy, particularly beneficial for non-verbal patient assessments.

Despite these advancements, challenges such as subtle pain expressions and such environmental factors as lighting variations remain, and further research is required to address these issues. To further enhance the effectiveness of pain level determination, future studies should expand the dataset to include a more diverse range of participants, encompassing various age groups, ethnic backgrounds, and medical conditions to improve robustness and generalizability. Additionally, refining weight indicators through extensive validation with medical professionals will ensure precise and clinically relevant estimations.

Integration of multimodal data, including physiological signals (e.g., heart rate variability, skin conductance) and vocal expressions, can offer more comprehensive pain analysis. Developing real-time automated pain detection systems for clinical environments, with user-friendly interfaces and minimal computational demands, should be pursued through pilot studies in hospitals and clinics. Conducting longitudinal studies will also be essential to understand pain expression dynamics, particularly in chronic pain conditions.

Furthermore, leveraging advanced machine learning techniques such as deep learning, reinforcement learning, and explainable AI approaches can enhance predictive accuracy and model transparency. Collaborative interdisciplinary research involving psychologists, neurologists, and pain specialists will further deepen our understanding of pain mechanisms, ultimately leading to improved patient outcomes and enhanced clinical pain management strategies.

In conclusion, this study lays important groundwork toward practical, reliable, and precise automated pain detection systems, with significant potential for improving patient care and pain management practices in diverse clinical settings.

## Figures and Tables

**Figure 1 jimaging-11-00151-f001:**
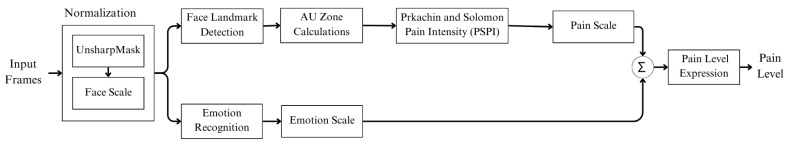
Pain level based on facial expressions estimated using a weighting method.

**Figure 2 jimaging-11-00151-f002:**
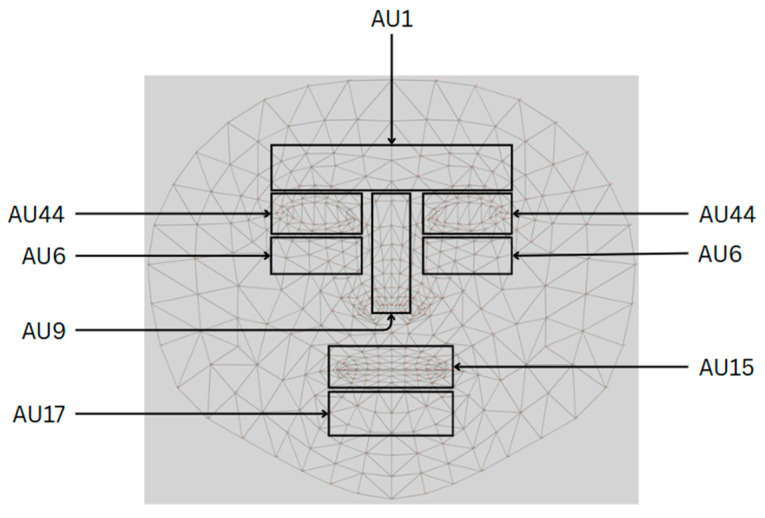
Facial features and their characteristics.

**Figure 3 jimaging-11-00151-f003:**
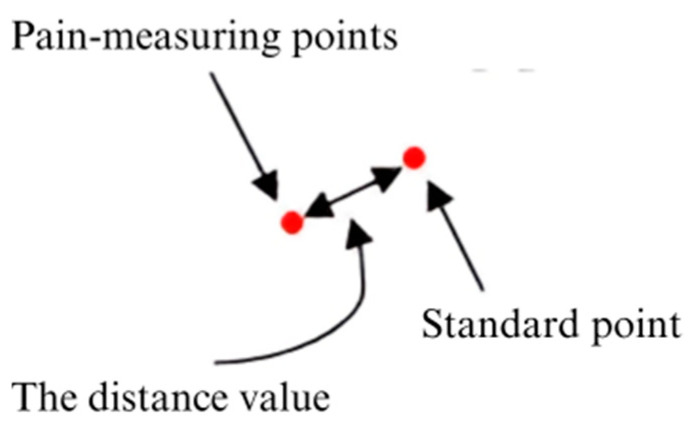
Model illustration of facial musculature changes corresponding to AUs.

**Figure 4 jimaging-11-00151-f004:**
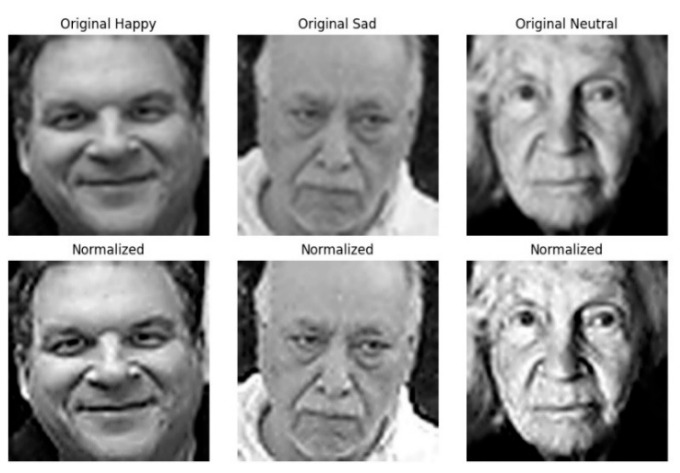
Original and normalized facial images illustrate the transformation process designed to ensure consistent pain assessment and facial expression analysis. The **top row** displays the original images depicting different facial expressions, including happiness, sadness, and neutrality, while the **bottom row** presents their normalized counterparts.

**Figure 5 jimaging-11-00151-f005:**
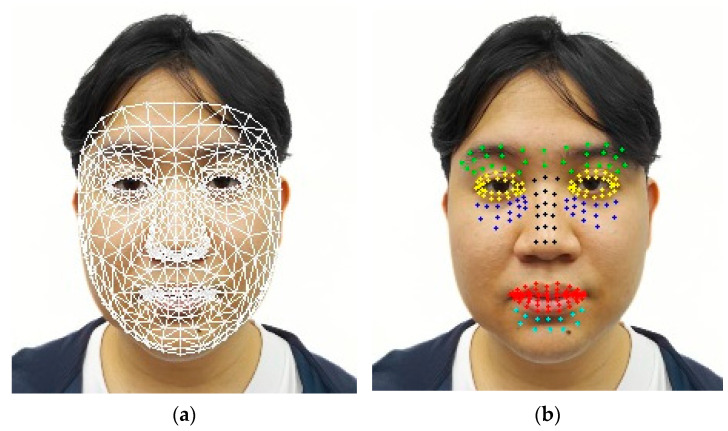
Example of facial landmark detection (**a**) alongside points inside each zone as examples (**b**).

**Figure 6 jimaging-11-00151-f006:**
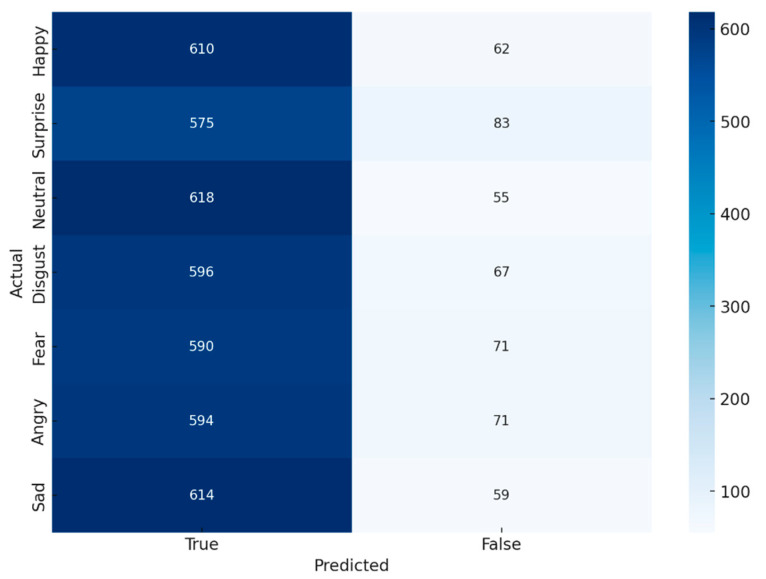
Confusion matrix for expression prediction.

**Figure 7 jimaging-11-00151-f007:**
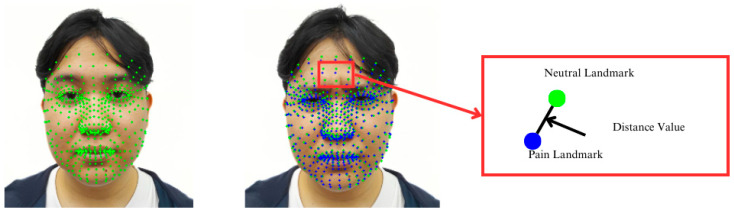
Example images of changes in facial muscle points.

**Figure 8 jimaging-11-00151-f008:**
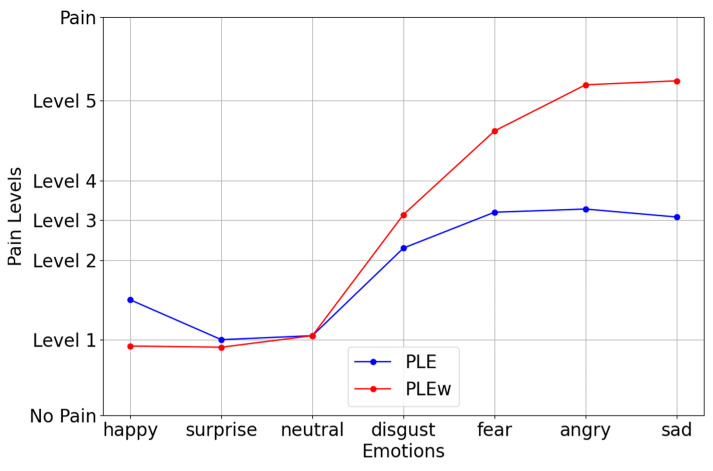
Linear graph of the average pain intensity by emotional category.

**Figure 9 jimaging-11-00151-f009:**
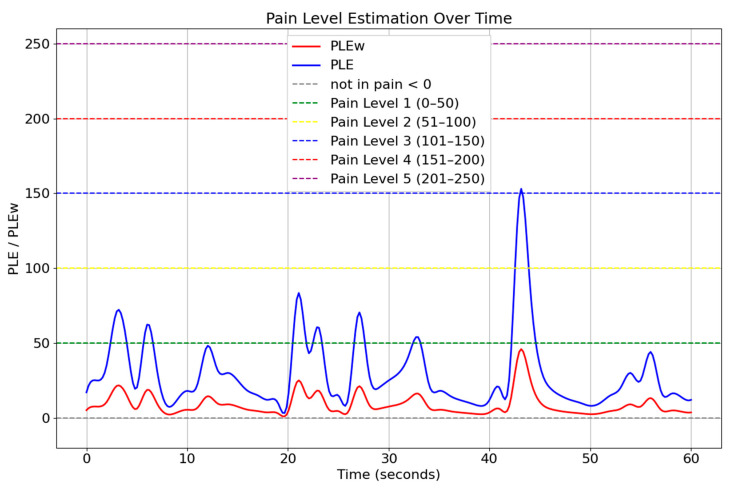
Temporal pain threshold profile for happiness (60 s interval).

**Figure 10 jimaging-11-00151-f010:**
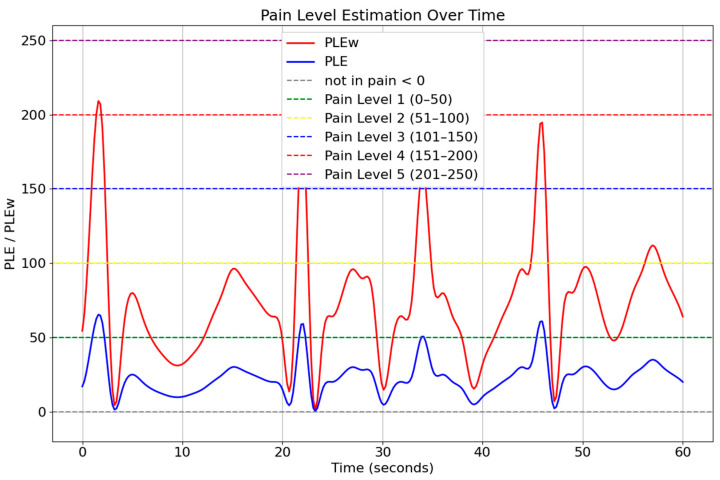
Temporal pain threshold profile for sadness (60 s interval).

**Table 1 jimaging-11-00151-t001:** Distribution of images corresponding to each facial expression category utilized in this study.

Expressions	Happiness	Surprise	Neutrality	Disgust	Fear	Anger	Sadness
Images	585	583	584	581	581	584	586

**Table 2 jimaging-11-00151-t002:** The table details the AU regions, their specific anatomical locations on the face, and the corresponding facial expressions associated with varying levels of pain and related emotional states.

AU	Description
AU1	Inner brow raiser zone, the space between the eyebrows
AU6	Cheek raiser zone, marks in the region under the eyes
AU9	Nose wrinkle zone, points where wrinkles appear on the nose
AU15	Lip corner depressor, patches near the mouth’s corners
AU17	Chin raiser zone, locations near the chin wrinkles
AU44	Squint zone, areas surrounding the squinting region

**Table 3 jimaging-11-00151-t003:** Facial expression illustrations with recognition confidence scores aid in pain assessment, as such expressions as fear, anger, and sadness often correlate with higher pain levels, while neutral or happy faces suggest lower discomfort. Integrating facial analysis improves accuracy by distinguishing pain-related features from non-pain-related ones.

Expressions	Happiness	Surprise	Neutrality	Disgust	Fear	Anger	Sadness
Image	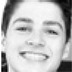	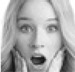	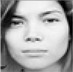	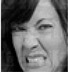	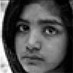	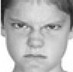	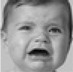
Probability	0.90	0.87	0.92	0.88	0.86	0.88	0.91

**Table 4 jimaging-11-00151-t004:** The confusion matrix summarizes the model’s performance in facial expression prediction, highlighting metrics such as precision, recall, accuracy, and F1-score. Results show that applying Unsharp Mask enhances prediction accuracy by improving facial expression differentiation.

Expressions	Precision	Recall	Accuracy	F1-Score	AUC	mAP
Happiness	0.9	0.9	0.88	0.9	0.9	0.9
Surprise	0.87	0.88	0.83	0.88	0.88	0.87
Neutrality	0.91	0.91	0.88	0.91	0.91	0.91
Disgust	0.89	0.9	0.86	0.9	0.9	0.89
Fear	0.89	0.89	0.85	0.89	0.89	0.89
Anger	0.89	0.9	0.86	0.89	0.89	0.89
Sadness	0.91	0.91	0.88	0.91	0.91	0.91

**Table 5 jimaging-11-00151-t005:** F1-score and precision results for pain threshold identification show the highest accuracy in neutral and sad expressions, while anger yields the lowest PLE, indicating detection challenges. The average scores confirm the model’s reliability in distinguishing pain-related facial cues across expressions.

Expressions	**Number of Images**	**PLE**	${PLE}_{w}$	${PLE}^{z[1–6]}$	${PLE}_{W}^{z[1–6]}$
Happiness	585	89.25	91.07	92.24	94.07
Surprise	583	85.35	89.19	90.17	92.60
Neutrality	584	87.48	88.66	90.02	91.10
Disgust	581	86.12	88.37	89.99	91.62
Fear	581	86.76	89.06	90.76	92.92
Anger	584	57.54	89.24	90.20	92.19
Sadness	586	89.11	92.09	92.64	94.53
Average	-	83.37	89.67	90.85	92.72

## Data Availability

Data are openly available in a public repository that issues datasets with KAGGLE. The data that support the findings of this study are available in the Face Expression Recognition dataset at https://www.kaggle.com/datasets/jonathanoheix/face-expression-recognition-dataset (accessed on 1 May 2024) [16].

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
