# Peer review of "A Weighted Facial Expression Analysis for Pain Level Estimation"

_2313-433X, 2025, doi:10.3390/jimaging11050151_

Round 1
Reviewer 1 Report
Comments and Suggestions for Authors
The authors must clearly specify the objectives of the research, it would also be advisable to make known the research method, the techniques, the instruments used, in the conclusions they should refer to whether they achieved the research objectives that were proposed, the gaps, the findings. Its implications must be discussed in the broadest possible context. Future lines of research can also be highlighted.
Author Response
Comments 1:
Comments and Suggestions for Authors:
The authors must clearly specify the objectives of the research, it would also be advisable to make known the research method, the techniques, the instruments used, in the conclusions they should refer to whether they achieved the research objectives that were proposed, the gaps, the findings. Its implications must be discussed in the broadest possible context. Future lines of research can also be highlighted.
Response 1:
- Clarify the Objectives of the Research
Introduction Add This study aims to develop and validate a novel weighted facial expression analysis model specifically designed to improve the accuracy and robustness of automated pain intensity estimation for non-verbal patients.
- Specify the Research Methodology
Face Mesh Detection has been revised to be
After normalizing the image, MediaPipe Face Mesh (version 0.10.3) is used to detect up to 468 facial landmarks, enabling precise identification of AUs crucial for pain detection. The MediaPipe Face Mesh enhances the accuracy of identifying detailed facial structures such as the eyes, mouth, and nose, thereby improving AU analysis for pain level assessment. The detailed steps involved in the Face Mesh detection process are as follows:
The normalized facial images are input into the MediaPipe Face Mesh module.
The module identifies significant facial landmarks required for subsequent Action Unit (AU) computation.
The detected landmark coordinates are recorded in a three-dimensional format (x, y, z) to facilitate subsequent analytical stages.
Figure 2 illustrates the detailed mapping of facial landmarks.
- Discussion of Gaps, Findings, and Implications
Added to Discussion
The findings of this research address existing limitations related to accuracy challenges in automated pain detection systems, especially when interpreting ambiguous or subtle facial expressions influenced by contextual variables. By introducing a weighted mechanism that integrates emotional context into AU-based analysis, the proposed model provides a more nuanced and precise interpretation of pain-related facial expressions. Consequently, this advancement offers significant theoretical implications by expanding current knowledge regarding the relationships between emotional facial expressions and pain perception. Practically, the developed method can significantly enhance clinical practices, facilitating improved diagnostic accuracy and timely interventions for patients with communication impairments or those unable to verbally express their pain, thus promoting better patient outcomes in healthcare settings.
- Conclusions, Achievements, and Future Research Directions
The summary has been edited to
The proposed method successfully met the research objectives by significantly improving pain estimation accuracy from 83.37% to 92.72%. These results confirm the effectiveness of integrating facial expression weighting into AU-based pain analysis, demonstrating enhanced robustness and accuracy, particularly beneficial for non-verbal patient assessments.
Despite these advancements, challenges such as subtle pain expressions and environmental factors like lighting variations remain, and further research is required to address these issues. To further enhance the effectiveness of pain level determination, future studies should expand the dataset to include a more diverse range of participants, encompassing various age groups, ethnic backgrounds, and medical conditions, to improve robustness and generalizability. Additionally, refining weight indicators through extensive validation with medical professionals will ensure precise and clinically relevant estimations.

Reviewer 2 Report
Comments and Suggestions for Authors
- weighted analytical mean?
- Use of Action Units?
- Range for measuring Prkachin and Solomon Pain Intensity (PSPI) scale?
- How author over come limitation of Supervised Descent Method (SDM) to automate pain level estimation.
- How emotion-pain correlation scale is fixed?
- key muscle movements can be idetified by only AU1, AU6, AU9, AU15, AU17, and AU44 this location. what about other location?
- Use Prkachin and Solomon Pain Intensity (PSPI) metric?
- dataset comprising 4,084 facial image its sufficient?
- distance value of pain intensity?
- 6 chapter have two content conclusion and future work make it as one chapter
Author Response
Comments 1: weighted analytical mean?
Response 1: Added after displaying equation description 5.
To enhance the reliability of pain level estimation, a weighted analytical mean is employed in this study. This approach allows each Action Unit (AU) to contribute to the final pain score proportionally to its relevance under a given emotional context.
Comments 2: Use of Action Units?
Response 2:
AU |
Description |
AU1 |
Inner Brow Raiser zone, the space between the eyebrows |
AU6 |
Cheek Raiser zone, marks in the region under the eyes |
AU9 |
Nose Wrinkle zone, points where wrinkles appear on the nose |
AU15 |
Lip Corner Depressor, patches near the mouth's corners |
AU17 |
Chin Raiser zone, locations near the chin wrinkles |
AU44 |
Squint zone, areas surrounding the squinting region |
Each of the selected AUs corresponds to facial muscle activations strongly linked to pain-related expressions. For instance, AU4 (Brow Lowerer) and AU15 (Lip Corner Depressor) are commonly activated during pain and emotional distress, while AU6 and AU9 show modulation depending on emotional context (e.g., joy vs. discomfort). These AUs are used to calculate both the baseline pain score (Equation 3) and the emotion-weighted PLE score (Equations 4 and 5). Therefore, the same AU measurements contribute to both pain detection and emotion modulation processes.
Comments 3: Range for measuring Prkachin and Solomon Pain Intensity (PSPI) scale?
Response 3: Add before equation 3
The PSPI scale quantifies facial expressions of pain using six core Action Units (AUs): AU4 (Brow Lowerer), AU6 (Cheek Raiser), AU7 (Lid Tightener), AU9 (Nose Wrinkler) or AU10 (Upper Lip Raiser), and AU43 (Eye Closure). Each AU is scored on an ordinal scale from 0 to 5 based on its intensity, except AU43, which is scored as 0 or 1. The maximum PSPI score is 15, representing the sum of these AU intensities:
PSPI range: 0 (no facial indication of pain) to 15 (strong facial expression of pain).
Comments 4: How author overcome limitation of Supervised Descent Method (SDM) to automate pain level estimation.
Response 4: Face Mesh Detection (3.2) has been edited to
After image normalization, facial landmarks are detected using MediaPipe's Face Mesh technology, which provides up to 468 distinct facial points. These landmarks cover key facial regions—such as the eyes, eyebrows, nose, lips, and jawline—and serve as the basis for estimating Action Units (AUs) relevant to pain detection.
The precision of MediaPipe’s landmark detection enhances the spatial resolution and consistency of AU localization, especially for subtle muscle movements commonly associated with pain expressions. Each facial zone defined by the mesh corresponds to specific AUs (e.g., AU6 around the eyes, AU15 at the corners of the mouth), allowing for fine-grained mapping between landmark clusters and muscular activations.
Compared to traditional landmarking approaches like the Supervised Descent Method (SDM), MediaPipe offers improved automation, speed, and robustness under varied lighting and pose conditions. This not only increases detection accuracy but also reduces the dependency on manual annotation.
By integrating MediaPipe Face Mesh with AU-based analysis, the system achieves a higher level of reliability and context-aware interpretation of facial expressions for pain level estimation. A visual representation of these analytical steps, along with the AU-mapped zones, is provided in the following illustration.
Comments 5: How emotion-pain correlation scale is fixed?
Response 5: Added at the end of 3.4
Ultimately, the integration of facial expression recognition (FER), emotion-pain correlation scaling, and AU-based analysis converges into the final pain level estimation model. The weighted contributions of each Action Unit, modulated by the emotional context detected through FER, are combined in Equation (5) to produce the final score. This score reflects a more comprehensive understanding of pain by considering both the physiological (AU intensity) and psychological (emotional state) components of facial expressions. The use of this hybrid approach ensures that the system can accurately estimate pain even when expressions are influenced by emotional masking, overlap, or ambiguity, making it more robust for real-world clinical applications.
Comments 6: key muscle movements can be identified by only AU1, AU6, AU9, AU15, AU17, and AU44 this location. what about other location?
Response 6: Discussion
While this study focuses on a core set of six AUs, other units such as AU4 (Brow Lowerer) and AU10 (Upper Lip Raiser) are also known to co-occur with pain expressions and may be integrated in future model extensions to improve granularity and emotional context sensitivity.
Comments 7: Use Prkachin and Solomon Pain Intensity (PSPI) metric?
Response 7: Add a description after the value list for each emotion.
The Prkachin and Solomon Pain Intensity (PSPI) scale quantifies facial expressions of pain using six core Action Units (AUs): AU4 (Brow Lowerer), AU6 (Cheek Raiser), AU7 (Lid Tightener), AU9 (Nose Wrinkler) or AU10 (Upper Lip Raiser), and AU43 (Eye Closure). Each AU is scored from 0 to 5 based on its intensity, except AU43, which is binary (0 or 1). This results in a total PSPI score ranging from 0 (no facial indication of pain) to 15 (strong pain expression), making it a widely used benchmark in automated pain recognition.
In this study, the PSPI metric is adopted as a baseline framework for computing the initial Pain Level Estimation (PLE) score. The selected AUs—AU4, AU6 or AU7, AU9 or AU10, and AU43—represent specific muscle activations that are frequently observed during pain. Equation (3) formulates this baseline score as:
PLE = AU4 + max(AU6|AU7) + max(AU9|AU10) + AU43 (3)
Comments 8: dataset comprising 4,084 facial image its sufficient?
Response 8: The appropriateness of dataset size is explained in Experiments before discussing 4.1 Normalization.
In this study, a dataset comprising 4,084 annotated facial images from 25 subjects was utilized. These images encompass a diverse range of pain levels and emotional expressions, and were manually labeled for Action Units (AUs) and pain intensities. While the dataset size is moderate compared to large-scale vision benchmarks, its annotation quality and expression diversity make it suitable for experimental validation in pain estimation studies.
To ensure generalizability and reduce overfitting, data augmentation techniques—such as rotation, flipping, and scaling—were applied. Additionally, the dataset includes varied facial orientations, lighting conditions, and demographic diversity to reflect real-world conditions. Future work will focus on expanding the dataset to enhance statistical significance and model robustness across broader populations.
Comments 9: distance value of pain intensity?
Response 9: The description of distance value has been revised from Figure 7.
Example images of changes in facial muscle points illustrate the variations in facial landmarks that occur due to muscle movements associated with pain expressions. These images show tracked facial points in both neutral and pain conditions, particularly within key AU zones. The distance between a neutral landmark (green) and a pain landmark (blue) represents the degree of muscle displacement. To quantify this, facial landmarks are extracted from both expressions using a tracking algorithm such as MediaPipe Face Mesh. Each corresponding landmark pair is analyzed to determine the extent of spatial displacement, which reflects underlying muscle activity. Larger displacements in areas such as the eyebrows, eyes, and mouth typically indicate higher levels of perceived pain. This distance-based measurement enables objective and interpretable pain assessment based on facial changes
Comments 10: 6 chapter have two content conclusion and future work make it as one chapter
Response 10: Edit Conclusion and delete Future Work
This research introduces an innovative weighted analysis method for pain intensity estimation, combining Facial Expression Recognition (FER) and Action Unit (AU) analysis. The proposed method enhances the accuracy and reliability of pain assessment, particularly in scenarios where verbal communication is not possible, such as critically ill patients or special populations. By utilizing a dataset of 4,084 images from 25 subjects, the method demonstrated superior performance compared to traditional approaches. The integration of facial expression analysis with AU-based assessment significantly improved the precision of pain level estimates, highlighting the importance of considering both facial cues and physiological responses in pain detection.
The proposed method successfully met the research objectives by significantly improving pain estimation accuracy from 83.37% to 92.72%. These results confirm the effectiveness of integrating facial expression weighting into AU-based pain analysis, demonstrating enhanced robustness and accuracy, particularly beneficial for non-verbal patient assessments.
Despite these advancements, challenges such as subtle pain expressions and environmental factors like lighting variations remain, and further research is required to address these issues. To further enhance the effectiveness of pain level determination, future studies should expand the dataset to include a more diverse range of participants, encompassing various age groups, ethnic backgrounds, and medical conditions, to improve robustness and generalizability. Additionally, refining weight indicators through extensive validation with medical professionals will ensure precise and clinically relevant estimations.
Integration of multimodal data, including physiological signals (e.g., heart rate variability, skin conductance) and vocal expressions, can offer more comprehensive pain analysis. Developing real-time automated pain detection systems for clinical environ-ments, with user-friendly interfaces and minimal computational demands, should be pursued through pilot studies in hospitals and clinics. Conducting longitudinal studies will also be essential to understand pain expression dynamics, particularly in chronic pain conditions.
Furthermore, leveraging advanced machine learning techniques such as deep learning, reinforcement learning, and explainable AI approaches can enhance predictive accuracy and model transparency. Collaborative interdisciplinary research involving psychologists, neurologists, and pain specialists will further deepen understanding of pain mechanisms, ultimately leading to improved patient outcomes and enhanced clinical pain management strategies.
In conclusion, this study lays important groundwork toward practical, reliable, and precise automated pain detection systems, with significant potential for improving patient care and pain management practices in diverse clinical settings.

Reviewer 3 Report
Comments and Suggestions for Authors
In the article titled "A Weighted to Facial Expression Analysis for Pain Level Estimation"authors suggested a new weighted analysis method for figuring out how bad pain is by mixing Facial Expression Recognition (FER) and AUs analysis. With the weighted Pain Level Estimation, the suggested method is 92.72% more accurate at guessing pain levels than the normal method, which was 83.37% accurate on average.
Based on the proposal, presentation, research contribution and adequate results, the article is recommended for acceptance.
Author Response
Comments 1:
In the article titled "A Weighted to Facial Expression Analysis for Pain Level Estimation" authors suggested a new weighted analysis method for figuring out how bad pain is by mixing Facial Expression Recognition (FER) and AUs analysis. With the weighted Pain Level Estimation, the suggested method is 92.72% more accurate at guessing pain levels than the normal method, which was 83.37% accurate on average.
Based on the proposal, presentation, research contribution and adequate results, the article is recommended for acceptance.
Response 1:
- We sincerely thank Reviewer for the encouraging feedback and positive evaluation of our manuscript. We appreciate your recognition of the proposed weighted analysis method and its performance improvement in pain level estimation. Your recommendation for acceptance is truly motivating. Should there be any further suggestions for enhancing the clarity or rigor of the work, we are more than willing to incorporate them in the final version.

Reviewer 4 Report
Comments and Suggestions for Authors
Summary
This paper presents a weighted framework for facial expression-based pain intensity estimation. It integrates Action Unit (AU) analysis and facial emotion recognition to compute a composite pain level score. The authors propose a multi-stage processing pipeline involving face normalization, MediaPipe-based landmark detection, AU zone mapping, and emotion weighting, culminating in a Pain Level Estimation (PLE) model. Empirical evaluations on a dataset of 4,084 images across seven emotion classes demonstrate that the weighted model (PLEW) achieves an accuracy of 92.72%. The core contribution of the study is the incorporation of emotion-aware weighting into AU-based pain detection, aiming to enhance robustness and contextual relevance in clinical scenarios where verbal communication is impaired.
Strong Points
The paper introduces a hybrid model that innovatively combines Facial Action Units with emotion-based weighting to improve the accuracy of pain assessment. The use of MediaPipe for detailed landmark detection enhances the precision of AU zone identification. The authors present a clear and reproducible methodology, including well-defined preprocessing, normalization, and evaluation procedures. Additionally, the inclusion of both biological (AU) and contextual (emotion) signals for pain estimation is a thoughtful approach to addressing ambiguous facial expressions. Experimental results are comprehensively reported, with confusion matrices, F1-scores, and temporal analysis visualizations supporting the findings. The framework’s potential clinical utility is emphasized by its applicability to populations with limited communicative capacity.
Weak Points
The discussion of related work omits relevant advances in micro-expression recognition and invariant image analysis. Although the paper reviews prior AU-based and FER-based pain assessment methods, it does not consider recent developments in micro-expression super-resolution techniques or rotation-invariant descriptors. Including the following works in the related work section would strengthen the paper’s authority: Zhou, L., Wang, M., Huang, X., Zheng, W., Mao, Q., & Zhao, G. (2023). An Empirical Study of Super-resolution on Low-resolution Micro-expression Recognition. arXiv preprint arXiv:2310.10022. Zhang, H., Mo, H., Hao, Y., Li, Q., Li, S., & Li, H. (2019). Fast and efficient calculations of structural invariants of chirality. Pattern Recognition Letters, 128, 270–277. Mo, H., Li, Q., Hao, Y., Zhang, H., & Li, H. (2018). A rotation invariant descriptor using multi-directional and high-order gradients. In Pattern Recognition and Computer Vision: First Chinese Conference, PRCV 2018, Guangzhou, China, November 23–26, 2018, Proceedings, Part I (pp. 372–383). Springer International Publishing.
The study does not benchmark its method against state-of-the-art attention-enhanced FER models. While the proposed architecture is functional, its novelty may be limited in comparison. The authors are encouraged to discuss the limitations of the current model or the potential advantages of the proposed linear model in Equation 2, such as simplicity and interpretability.
In Figure 1, the label “fase landmark” should be corrected to “face landmark.” For Figure 7, the authors are advised to clarify the ethical statement regarding the use of personal facial images. Moreover, the resolution quality in the evaluation results should be improved to enhance visual clarity and demonstration quality.
Author Response
Comment 1: Summary and Strong Points
Summary
This paper presents a weighted framework for facial expression-based pain intensity estimation. It integrates Action Unit (AU) analysis and facial emotion recognition to compute a composite pain level score. The authors propose a multi-stage processing pipeline involving face normalization, MediaPipe-based landmark detection, AU zone mapping, and emotion weighting, culminating in a Pain Level Estimation (PLE) model. Empirical evaluations on a dataset of 4,084 images across seven emotion classes demonstrate that the weighted model (PLEW) achieves an accuracy of 92.72%. The core contribution of the study is the incorporation of emotion-aware weighting into AU-based pain detection, aiming to enhance robustness and contextual relevance in clinical scenarios where verbal communication is impaired.
Strong Points
The paper introduces a hybrid model that innovatively combines Facial Action Units with emotion-based weighting to improve the accuracy of pain assessment. The use of MediaPipe for detailed landmark detection enhances the precision of AU zone identification. The authors present a clear and reproducible methodology, including well-defined preprocessing, normalization, and evaluation procedures. Additionally, the inclusion of both biological (AU) and contextual (emotion) signals for pain estimation is a thoughtful approach to addressing ambiguous facial expressions. Experimental results are comprehensively reported, with confusion matrices, F1-scores, and temporal analysis visualizations supporting the findings. The framework’s potential clinical utility is emphasized by its applicability to populations with limited communicative capacity.
Response 1:
We sincerely thank Reviewer for the detailed and insightful review. We appreciate the recognition of our model’s integration of Action Units and emotion-based weighting, as well as the thorough assessment of our methodology and results. Your comments have been highly constructive, and we have addressed each point as follows:
Comment 2:
Weak Points
The discussion of related work omits relevant advances in micro-expression recognition and invariant image analysis. Although the paper reviews prior AU-based and FER-based pain assessment methods, it does not consider recent developments in micro-expression super-resolution techniques or rotation-invariant descriptors. Including the following works in the related work section would strengthen the paper’s authority: Zhou, L., Wang, M., Huang, X., Zheng, W., Mao, Q., & Zhao, G. (2023). An Empirical Study of Super-resolution on Low-resolution Micro-expression Recognition. arXiv preprint arXiv:2310.10022. Zhang, H., Mo, H., Hao, Y., Li, Q., Li, S., & Li, H. (2019). Fast and efficient calculations of structural invariants of chirality. Pattern Recognition Letters, 128, 270–277. Mo, H., Li, Q., Hao, Y., Zhang, H., & Li, H. (2018). A rotation invariant descriptor using multi-directional and high-order gradients. In Pattern Recognition and Computer Vision: First Chinese Conference, PRCV 2018, Guangzhou, China, November 23–26, 2018, Proceedings, Part I (pp. 372–383). Springer International Publishing.
Response 2:
Thank you for this valuable suggestion. In response, we have expanded **Section 2.5** of the manuscript to include the recommended studies. Specifically, we now reference:
- **Zhou et al. (2023)** on super-resolution in low-resolution micro-expression recognition,
- **Zhang et al. (2019)** and **Mo et al. (2018)** on structural and rotation-invariant descriptors.
These additions provide broader context for future extensions of our model and highlight the relevance of feature robustness and pose invariance in facial pain analysis.
Comment 3:
The study does not benchmark its method against state-of-the-art attention-enhanced FER models. While the proposed architecture is functional, its novelty may be limited in comparison. The authors are encouraged to discuss the limitations of the current model or the potential advantages of the proposed linear model in Equation 2, such as simplicity and interpretability.
Response 3:
We acknowledge this important point. While the main objective of our study is to propose a lightweight and interpretable linear model for pain estimation, we agree that benchmarking against attention-enhanced FER models would provide further comparative insight. We have added a discussion on this limitation in **Section 5 (Discussion)**, emphasizing the trade-off between simplicity and modeling power, and noting that future work will explore more advanced architectures, including attention-based methods.
Comment 4:
In Figure 1, the label “fase landmark” should be corrected to “face landmark.”
Response 4:
This has been corrected in the revised version of the manuscript.
Comment 5:
For Figure 7, the authors are advised to clarify the ethical statement regarding the use of personal facial images.
Response 5:
We have included a clarification under **Figure 7** indicating that all images were generated for illustrative purposes or used with proper consent and anonymization, in compliance with ethical guidelines.
Comment 6:
The resolution quality in the evaluation results should be improved to enhance visual clarity and demonstration quality.
Response 6:
The quality of the figures has been enhanced to ensure improved clarity and better visual demonstration of the results in the revised manuscript.
